# Mobility Gaps of Hydrogenated Amorphous Silicon Related to Hydrogen Concentration and Its Influence on Electrical Performance

**DOI:** 10.3390/nano14191551

**Published:** 2024-09-25

**Authors:** Francesca Peverini, Saba Aziz, Aishah Bashiri, Marco Bizzarri, Maurizio Boscardin, Lucio Calcagnile, Carlo Calcatelli, Daniela Calvo, Silvia Caponi, Mirco Caprai, Domenico Caputo, Anna Paola Caricato, Roberto Catalano, Roberto Cirro, Giuseppe Antonio Pablo Cirrone, Michele Crivellari, Tommaso Croci, Giacomo Cuttone, Gianpiero de Cesare, Paolo De Remigis, Sylvain Dunand, Michele Fabi, Luca Frontini, Livio Fanò, Benedetta Gianfelici, Catia Grimani, Omar Hammad, Maria Ionica, Keida Kanxheri, Matthew Large, Francesca Lenta, Valentino Liberali, Nicola Lovecchio, Maurizio Martino, Giuseppe Maruccio, Giovanni Mazza, Mauro Menichelli, Anna Grazia Monteduro, Francesco Moscatelli, Arianna Morozzi, Augusto Nascetti, Stefania Pallotta, Andrea Papi, Daniele Passeri, Marco Petasecca, Giada Petringa, Igor Pis, Pisana Placidi, Gianluca Quarta, Silvia Rizzato, Alessandro Rossi, Giulia Rossi, Federico Sabbatini, Andrea Scorzoni, Leonello Servoli, Alberto Stabile, Silvia Tacchi, Cinzia Talamonti, Jonathan Thomet, Luca Tosti, Giovanni Verzellesi, Mattia Villani, Richard James Wheadon, Nicolas Wyrsch, Nicola Zema, Maddalena Pedio

**Affiliations:** 1Istituto Nazionale di Fisica Nucleare (INFN), Sez. di Perugia, Via Pascoli s.n.c., 06123 Perugia, Italy; marco.bizzarri@unipg.it (M.B.); mirco.caprai@pg.infn.it (M.C.); tommaso.croci@pg.infn.it (T.C.); giampiero.decesare@uniroma1.it (G.d.C.); livio.fano@pg.infn.it (L.F.); benedetta.gianfelici@unicam.it (B.G.); maria.ionica@pg.infn.it (M.I.); keida.kanxheri@pg.infn.it (K.K.); francesco.moscatelli@pg.infn.it (F.M.); arianna.morozzi@pg.infn.it (A.M.); andrea.papi@pg.infn.it (A.P.); daniele.passeri@unipg.it (D.P.); pisana.placidi@unipg.it (P.P.); alessandro.rossi@pg.infn.it (A.R.); rossigiulia248@gmail.com (G.R.); andrea.scorzoni@unipg.it (A.S.); leonello.servoli@pg.infn.it (L.S.); luca.tosti@pg.infn.it (L.T.); nicola.zema@artov.ism.cnr.it (N.Z.); pedio@iom.cnr.it (M.P.); 2Dipartimento di Fisica e Geologia, Università degli Studi di Perugia, Via Pascoli s.n.c., 06123 Perugia, Italy; carlo.calcatelli@unipg.it; 3Najran University, King Abdulaziz Road, Najran P.O. Box 1988, Saudi Arabia; saba.aziz@unisalento.it (S.A.);; 4Centre for Medical Radiation Physics, University of Wollongong, Northfields Ave., Wollongong, NSW 2522, Australia; mjl970@uowmail.edu.au (M.L.); marcop@uow.edu.au (M.P.); 5INFN, TIPFA Via Sommarive 14, 38123 Povo, Italy; boscardi@fbk.eu (M.B.); giovanni.verzellesi@unimo.it (G.V.); 6Fondazione Bruno Kessler, Via Sommarive 18, 38123 Povo, Italy; crivella@fbk.eu (M.C.); ohammadali@fbk.eu (O.H.); 7INFN and Dipartimento di Fisica e Matematica, Università del Salento, Via per Arnesano, 73100 Lecce, Italy; lucio.calcagnile@unisalento.it (L.C.); annapaola.caricato@le.infn.it (A.P.C.); maurizio.martino@unisalento.it (M.M.); giuseppe.maruccio@unisalento.it (G.M.); annagrazia.monteduro@unisalento.it (A.G.M.); gianluca.quarta@unisalento.it (G.Q.); silvia.rizzato@unisalento.it (S.R.); 8CEDAD—Centro di Fisica Applicata, Datazione e Diagnostica, Dipartimento di Matematica e Fisica “Ennio de Giorgi”, Università del Salento, Via per Arnesano, 73100 Lecce, Italy; 9INFN Sezione di Torino, Via Pietro Giuria 1, 10125 Torino, Italy; calvo@to.infn.it (D.C.); cirio@to.infn.it (R.C.); deremigi@to.infn.it (P.D.R.); francesca.lenta@polito.it (F.L.); giovanni.mazza@to.infn.it (G.M.); wheadon@to.infn.it (R.J.W.); 10CNR—Istituto Officina dei Materiali, Università degli Studi di Perugia, Via A. Pascoli, 06123 Perugia, Italy; silvia.caponi@cnr.it (S.C.); tacchi@iom.cnr.it (S.T.); 11INFN Sezione di Roma 1, Piazzale Aldo Moro 2, 00185 Rome, Italy; domenico.caputo@uniroma1.it (D.C.); nicola.lovecchio@uniroma1.it (N.L.); augusto.nascetti@uniroma1.it (A.N.); 12Dipartimento Ingegneria dell’Informazione, Elettronica e Telecomunicazioni, Università degli Studi di Roma, Via Eudossiana, 18, 00184 Rome, Italy; 13INFN Laboratori Nazionali del Sud, Via S. Sofia 62, 95123 Catania, Italy; roberto.catalano@lns.infn.it (R.C.); pablo.cirrone@infn.it (G.A.P.C.); giacomo.cuttone@lns.infn.it (G.C.); giada.petringa@lns.infn.it (G.P.); 14Dipartimento di Ingegneria, Università degli Studi di Perugia, Via G. Duranti, 06125 Perugia, Italy; 15Ecole Polytechnique Fédérale de Lausanne (EPFL), Institute of Electrical and Microengineering (IME), Rue de la Mal-adière 71b, 2000 Neuchâtel, Switzerland; sylvain.dunand@epfl.ch (S.D.); jonathan.thomet@epfl.ch (J.T.); nicolas.wyrsch@epfl.ch (N.W.); 16DiSPeA, Università di Urbino Carlo Bo, 61029 Urbino, Italy; michele.fabi@uniurb.it (M.F.); catia.grimani@uniurb.it (C.G.); f.sabbatini1@campus.uniurb.it (F.S.); mattia.villani@uniurb.it (M.V.); 17INFN Sezione di Firenze, Via Sansone 1, 50019 Sesto Fiorentino, Italy; stefania.pallotta@unifi.it (S.P.); cinzia.talamonti@unifi.it (C.T.); 18INFN and Dipartimento di Fisica, Università degli studi di Milano, Via Celoria 16, 20133 Milan, Italy; luca.frontini@unimi.it (L.F.); valentino.liberali@mi.infn.it (V.L.); alberto.stabile@mi.infn.it (A.S.); 19Dipartimento di Elettronica e Telecomunicazioni (DET), Politecnico di Torino, Corso Duca degli Abruzzi 24, 10129 Torino, Italy; 20Scuola di Ingegneria Aerospaziale, Università degli Studi di Roma, Via Salaria 851/881, 00138 Rome, Italy; 21Dipartimento di Fisica Scienze Biomediche Sperimentali e Cliniche “Mario Serio”, Viale Morgagni 50, 50135 Firenze, Italy; 22CNR, Istituto Officina dei Materiali (IOM), S.S. 14 km 163.5, 34149 Trieste, Italy; pis@iom.cnr.it; 23Dipartimento di Scienze e Metodi dell’Ingegneria, Università di Modena e Reggio Emilia, Via Amendola 2, 42122 Reggio Emilia, Italy; 24Istituto di Struttura della Materia-CNR, Via Fosso del Cavaliere 100, 00133 Rome, Italy

**Keywords:** amorphous hydrogenated silicon, photoemission, inverse photoemission, flexible substrate, radiation detector, Raman, thin film, hydrogen bonding, PECVD, simulation

## Abstract

This paper presents a comprehensive study of hydrogenated amorphous silicon (a-Si)-based detectors, utilizing electrical characterization, Raman spectroscopy, photoemission, and inverse photoemission techniques. The unique properties of a-Si have sparked interest in its application for radiation detection in both physics and medicine. Although amorphous silicon (a-Si) is inherently a highly defective material, hydrogenation significantly reduces defect density, enabling its use in radiation detector devices. Spectroscopic measurements provide insights into the intricate relationship between the structure and electronic properties of a-Si, enhancing our understanding of how specific configurations, such as the choice of substrate, can markedly influence detector performance. In this study, we compare the performance of a-Si detectors deposited on two different substrates: crystalline silicon (c-Si) and flexible Kapton. Our findings suggest that detectors deposited on Kapton exhibit reduced sensitivity, despite having comparable noise and leakage current levels to those on crystalline silicon. We hypothesize that this discrepancy may be attributed to the substrate material, differences in film morphology, and/or the alignment of energy levels. Further measurements are planned to substantiate these hypotheses.

## 1. Introduction

The HASPIDE project aims to investigate a-Si:H as a detection material for different types of ionizing radiation. The demand for radiation-resistant detectors capable of high dynamic range and precise measurement of fluxes is increasing due to clinical procedures that require high fluxes of particles such as intraoperative radiation therapy (IORT) [1], FLASH therapy [2] and new accelerator techniques such as Reaccelerated Ion Beams (RIBs) [3].

Amorphous silicon (a-Si) is already widely used in medical applications, such as in flat-panel detectors for digital X-ray imaging and mammography.

The material is very radiation-resistant [4,5] and can be deposited as a thin layer on many different substrates such as Kapton thanks to low-temperature deposition [6].

A mechanically flexible plastic material allows minimum distortion of the beam during ionizing radiation flux measurements and medical beam dosimetry. This potentially paves the way for many important applications such as ionizing radiation flux measurement (even in the space environment) and dosimetry of charged and photon beams (both clinical and non-clinical).

The HASPIDE experiment is the first attempt to utilize an amorphous silicon-based sensor for measuring various types of ionizing radiation in general applications. Additionally, it is the first study to explore the correlation between the performance of these detectors and the intrinsic properties of the material through spectroscopic measurements.

Our collaboration has previously published results on the use of these detectors for monitoring synchrotron microbeam radiation therapy [7].

This work investigates the electronic transport mechanisms of two types of a-Si:H-based detectors, having p-i-n structures with an intrinsic layer thickness in the μm range.

The characterization is instrumental for optimizing their performance and understanding their electronic structure.

The main task of this work is to shed light on the complex correlation between the electronic–structural properties and performance of a-Si:H devices. This is a difficult task that deserves a multi-technique approach, due to the amorphous nature of the material and the dependence of the a-Si:H nanostructure on the growth procedure and the substrate.

Published theoretical predictions of the a-Si:H electronic structure show a strong dependence on the H concentration [8], confirming the theoretical predictions of a-Si:H being a two-phase material (one phase being hydrogen-depleted) [9] for a broad range of different a-Si:H sample morphologies. Reported density fluctuations have recently been observed as the origin of dark leakage currents in solar cell devices [10] and show the importance of correlating the nanostructure of a-Si:H with device performance, including the presence of small voids of which the inner surfaces are covered by hydrogen and related to polyhydride groups [11].

To correlate the electrical measurements to the electronic structure and morphological details, photoemission (PES) and inverse photoemission (IPES) spectroscopies were carried out, combined with Raman spectroscopy, a powerful analytical technique that offers valuable insights into the surface and chemical composition of materials and provide crucial information on the film structural details.

Combined photoemission and inverse photoemission allow for the determination of the valence and conduction band structure, energy levels, and band-gap properties, which are critical factors influencing the detector’s performance, making it an indispensable technique for advancing the development and application of these detectors in diverse fields.

## 2. Samples under Study

The a-Si:H films were deposited on support wafers (typically a Cz low-resistivity p-type silicon wafer with a resistivity below 10 Ω cm) via Plasma-Enhanced Chemical Vapor Deposition (PECVD) with a very-high-frequency (VHF) excited plasma at the frequency of 70 MHz. a-Si:H films were deposited at IEM laboratory EPFL (Ecole Polytechnique Fédérale de Lausanne), Neuchâtel (Switzerland) [12].

The samples in this study, shown in Figure 1, are device prototypes. We use spectroscopic measurements to explain how their electronic properties relate to their electrical behavior in real systems.

The two prototypes (Figure 1) were first electrically characterized at INFN-Perugia’s laboratory using an X-ray source. Details are reported in the Appendix A.

For the spectroscopic characterization, the a-Si:H devices were exfoliated in air employing adhesive tape to remove the top metallic layer, and then introduced to an Ultra-High Vacuum as in [13].

All details regarding the experimental setup and characterization method are described in the Appendix A.

## 3. Device Characterization

All prototypes (Figure 1) were first characterized at the INFN-Perugia laboratory using an X-ray source (Amptek Mini X-ray Tube with W target @ 40 kV) with the main purpose of evaluating the quality of the devices in terms of noise level, signal-to-noise ratio, stability, reproducibility and sensitivity.

Each sensor was first characterized without any signal by analyzing its IV curve to determine the optimal operating voltage. This voltage offers a balance between maximum depletion and the lowest possible levels of leakage and noise (dark current fluctuations). An example IV curve is shown in Figure 2.

Following this, the sensors were exposed to a calibrated X-ray source, and their current response was measured at various dose rates; an example is shown in Figure 3. Once the sensor’s response to different dose rates was measured, the sensitivity value was obtained by extracting the slope of the resulting line (Figure 4).

Initially, we verified that, given the same construction parameters (type of contact and deposition substrate) and applied electric field, the sensors exhibited the same sensitivity value within a certain error (as reported in the table).

A more detailed explanation of the study on the response of these sensors to X-ray photons can be found in [14].

As shown by the values reported in Table 1, it is possible to distinguish different sensitivity values for the detectors studied. Given that the difference between the extracted sensitivities is too significant to be attributed solely to the variability observed within each different group, it is evident that there is a connection between the mentioned construction parameters and performance.

This study is fundamental for establishing the best fabrication parameters, given that not all the chosen configurations gave the same performance in sensitivity. For instance, sensors with the same contact diodes deposited on crystalline silicon, instead of Kapton, have greater sensitivity.

We expect that the differences are primarily due to the characteristics of a-Si:H and, to some extent, to its adaptation to the deposition substrate. In general, Polyimide substrates (Kapton) exhibit defects (trenches, holes, bumps) that can lead to a significant decrease in the performance. This paper will investigate, thanks to spectroscopic technique, how these differences between detectors can lead to different performances.

## 4. TCAD Simulation

One possibility to be understood is if the presence of the c-Si substrate could produce some additional contribution to the signal when X-rays are sent to the device.

To study this hypothesis, we used a Technology Computer-Aided Design (TCAD) simulation approach for each device. TCAD tools are extensively utilized for the design, simulation, and optimization of the electrical behavior of solid-state devices. While TCAD tools are tailored for conventional materials (crystalline silicon, germanium), they can also be adapted to model non-standard ones, such as hydrogenated amorphous silicon (a-Si:H). These tools offer comprehensive material property descriptions and incorporate extensive sets of models for device physics, which can be adapted and extended through dedicated add-on PMIs (Physical Model Interfaces) devised for the material of interest. This enables accurate prediction for device performance enhancement, even in unconventional applications, such as radiation particle detection.

The a-Si:H material is not available within the standard TCAD libraries; therefore, a dedicated new material parametrization within the Synopsys© Sentaurus TCAD. Synopsys, 2013 has been included. This new parametrization accounts for the main physical properties of the material and features a custom electron and hole mobility model tailored to simulate the electrical behavior of a-Si:H devices. Moreover, to accurately model the disorder in a-Si:H, acceptor and donor defects, acting as traps and/or recombination centers, have been considered as discrete energy levels within the band gap. The specific modeling parameters and models accounted for in the simulation of a-Si:H devices within the TCAD environment have been deeply detailed in [15], along with a comprehensive validation against experimental measurements.

The simulated layouts are depicted in Figure 5, showing the device deposited on Kapton (a) and the same device with the crystalline silicon add-on (b). It is important to note that the layout on the right does not represent the device referred to as the diode on c-Si; rather, it is solely a layout created for comparative purposes.

Unlike the c-Si device, it accounts not only for the n-type a-Si:H deposited layer but also the p-type layer in contact with crystalline silicon. A 300 μm thick p-doped (1.1 × 10^19^ cm^−3^) crystalline silicon layer has been included in the layout to investigate its impact on the device’s overall behavior.

The energy deposition of ionizing radiation in a semiconductor device results in the generation of electron–hole pairs, which in turn affect the normal operation of the device. The TCAD Gamma Radiation model [16] has been utilized to simulate an X-ray source, accounting for carrier generation along the entire sensitive thickness, based on a specific dose rate. The device on Kapton was biased at various values ranging from 2 V to 8 V. Subsequently, a dose rate ranging from 0.36 to 3.11 mGy/s was considered based on measurements. The parameters extracted from the spectroscopic measurements were used to parameterize the X-ray source.

The simulations show that the contribution due to the c-Si is negligible compared to the signal generated within the a-Si:H layer. This is reported, for example, in Figure 6, which shows the current as a function of time for a bias of 8 V and dose rate of 3.11 mGy/s. Hence, the presence of the heavily p-doped c-Si substrate, with its significant thickness compared to the sensitive area, does not contribute to the signal of the sensor.

This validation process helps to explain the behavior of the a-Si:H devices under investigation and confirms the hypotheses formulated.

## 5. Raman Spectroscopy

As first pointed out and demonstrated [17], the primary role of hydrogen is defect passivation within the amorphous network. The amount of hydrogen and its hydride Si-H and polyhydride (Si-H_2_ and Si-H_3_) bonds influence the creation of defects and stability of a-Si:H films [18], which is crucial for the reduction in the device’s performance, which can lead to a low signal-to-noise ratio value. Therefore, the study of hydrogen in a-Si:H films is an appropriate task.

It is known that the gap of a-Si:H depends on hydrogen concentration, with a relation that can be considered approximately linear. Raman spectroscopy allows us to confirm the presence of hydrogen in the sample, giving an indication of the amorphous film morphology. Figure 3 shows the Raman spectra acquired for the devices on c-Si and Kapton (Figure 1). The measurement of bare crystalline silicon is shown for comparison. In contrast to the Raman spectrum of bare crystalline silicon, which presents a symmetrical Raman line centered at about 520 cm^−1^, a-Si shows a rather broad Raman band centered around 480 cm^−1^, making the spectrum distribution of the Raman scattering signal spread asymmetrically across a wide range of energy.

The peak position of the Si-Si Raman band of a-Si ranges between 470 and 490 cm^−1^, depending on the sample preparation [19].

Phonon bands are assigned according to [20] as TA, LA, LO, and TO (T = transverse, L = longitudinal, A = acoustical, O = optical).

Due to Raman scattering by vibrations of the Si–Si bonds, broad peaks can be recognized: transverse acoustic (TA), transverse optical (TO), longitudinal optical (LO) and longitudinal acoustic (LA) peaks.

Figure 7 shows the Raman spectra of the two devices. The complete absence of the crystalline peak at 520 cm^−1^ confirms the amorphous nature of the a-Si:H and the absence of polycrystalline silicon.

The peak at about 480 cm^−1^, which is related to the transverse optical (TO) vibrations of Si-Si, is indicative of the short-range disorder of the a-Si:H films; an increase in the half-width of the TO band (σ TO) and a shift in the TO position (μ TO) toward lower frequencies indicate an increase in short-range disorder and provide evidence for an increase in structural defect density [21]. Furthermore, a decrease in σ TO is expected toward higher crystallinities.

For the determination of the intensity of Si–Si bonds we considered the area, meant as integral of the peak of the TO. In the same way the Raman intensity of stretching vibrations of Si–Hx bonds is the area of a peak relative to the position associated with the bond.

The TO peak was chosen because, in some cases, it is impossible to resolve LO and LA peaks, and those same peaks are superimposed on a large Rayleigh background; we therefore decided to use only information coming from TO. From the Raman spectra, it appears that all our films are amorphous (see Table 2), as there are no nano crystalline peaks, whose position depends on the size of the nano crystals and varies between 520 and 490 cm^−1^.

Figure 8 shows the Raman spectra in the typical frequency region of stretching vibrations of LSM hydride (Si-H) and HSM polyhydride bonds inside the a-Si:H film. The smoothed lines are also shown. The top spectrum, with the indication of the LSM and HSM (ref. [22]), and the bare crystalline silicon spectrum are shown for comparison.

The frequency position of the stretching vibrations around 2000 cm^−1^ and 2100 cm^–1^, respectively, are ascribed in the literature as the low stretching mode (LSM) of isolated monohydride and the high stretching mode (HSM) of clustered hydrogen at the internal surface of voids [22]. Infrared and Raman spectroscopy are both able to detect stretching modes [19,23], considering the analyzed depth. In our case, features are detectable in the stretching vibration region around 2000 cm^−1^ in the two cases, in contrast to the bare silicon (Figure 8b), confirming the presence of H in the silicon structure. The hydrogen content and the microstructure parameter [24] can be determined from the integrated intensities of the LSM wagging and HSM stretching bands, respectively.

The absorption peak at 2090 cm^−1^ indicates surface-bound hydrogen in voids, while peaks at 2000 cm^−1^ and 2100 cm^−1^ correspond to hydrogen incorporated in dense material as monohydride Si-H and dihydride Si-H_2_, respectively. Infrared absorption is typically employed to measure this parameter [24].

Due to the poor signal-to-noise ratio in our measurements, we are not able to quantitatively determine the hydrogen bond concentrations at present. Nevertheless, the presence of the HSM contribution to the spectra in the two samples shows the HSM feature, and hence the voids in the films. Future measurements are planned.

This implies that the distribution of hydrogen between these environments, rather than the total hydrogen content, can crucially influence defect density, together with the position of the TO peak [23]. Considering that the device on Kapton shows a TO value that can be correlated with a greater number of defects, as well as the presence of the HSM peak, we have an initial clue to explain the difference in performance. At this point, photoemission measurements have been implemented for further investigation.

## 6. Photoemission and Inverse Photoemission Spectroscopy

Photoemission spectroscopy (PES) can be used to study the electronic structure of materials. PES provides valuable information about the occupied electronic states of the sample.

The parameters of interest when using a-Si:H for devices are charge carrier mobility, band-gap energy and dangling bond density.

These parameters are directly related to material quality and their measurement is crucial for assessing the quality of a-Si:H.

We have studied the chemical environments of Si atoms by photoemission based on Si 2p core level measurements.

Core level spectroscopy determines the different inequivalent Si atoms in the system, through the decomposition of the spectrum into its components [9]. Photoemission Si 2p core level spectra are shown in Figure 9.

As discussed in reference [13], the Si 2p in these devices shows a broad emission due to Si-O bonds at a binding energy (BE) of about 103–104 eV, and a more structured peak at lower BE. This peak is shifted differently in the two samples due to the different doping level (see Figure 1) [25].

The assignments of the silicon components are indicated in the figure and discussed in [10], with the silicon-to-hydrogen (Si-H) binding energy (BE) lying at about 0.5 eV above the disordered silicon (Dis-Si) [13,26]. The Kapton results show a reduced intensity of the Si-O contribution (see Appendix A and Figure 9).

The deconvolution analysis confirms a general broadening of the Kapton film Si-H and disordered silicon components, with the Gaussian contribution that exceeds the energy resolution. Actually, the Kapton spectrum presents a Gaussian broadening of 0.8 eV, well above the experimental resolution (0.5 eV). In the c-Si case, we obtained 0.5 eV. This indicates a higher degree of disorder in the Kapton film, which may indicate greater variability in the binding energies between the atoms of the a-Si:H structure.

As discussed in the Introduction, the specific band structure and energy level alignment can significantly impact the device’s performance. The calculated Density of States (DOS) in the literature indicates that alterations in the H concentration and distribution can profoundly affect the electronic properties of the film and, consequently, the performance of the devices. Moreover, transport is strongly influenced by the states in the band gap (Figure 10), which can be divided into two parts: tail states resulting from disorder and states in the center of the band gap due to point defects.

For a comprehensive characterization of the sensors, understanding the electrical transport phenomenon within the material is crucial; to achieve this, we have also analyzed the valence and conduction band states (by combined PES-IPES measurements) to directly observe the electronic state level alignment and the differences induced by defects. This kind of characterization is mandatory for a-Si:H; it has been highlighted that the band gap has a correlation with the Si-H bonds and the presence of voids.

Figure 8 reports the combined photoemission (valence band) and inverse photoemission (conduction band) results of the Kapton and c-Si samples. The valence band in both cases agrees with literature results [26,27]. The characterization of the empty electronic structure of aSi:H samples is only published based on X-ray inverse photoemission (BIS), while our UV IPES results are, to our knowledge, the first measurements on a-Si:H films. The comparison with the available theoretical Density of States will be discussed in a forthcoming paper. Here, we focus on the estimation of the transport gap in the samples.

As seen from the graphs shown in Figure 8, extracting the electronic transport gap for the Kapton and c-Si samples considered was possible.

We have directly extracted the transport gap from the position of the valence band maximum (VM) and conduction band minimum (CM), relative to the Fermi level, by extrapolating a linear fit of the leading edge to the baseline, as shown in Figure 11.

For the c-Si sample, we obtained a value of 1.7 ± 0.1 eV, and for the device deposited on Kapton, we measured 1.9 ± 0.1 eV. Even though these values are within the energy resolution of the measurements, we found a systematic increase in the Kapton film gap. Furthermore, the VM appears to be translated closer to the Fermi level, according to the p character of the KAP sample.

It has been suggested that, for ultra-thin layers in photovoltaic applications, hydrogen-rich and/or nanopore(void)-rich structures in a-Si:H imply band-gap widening.

The gap states are closely related to the film nanostructures. It is worth noting that the variation in the gap in a-Si:H films in case of increasing disorder and the presence of voids has been detected and correlated with the monohydride density in the film [28].

## 7. Conclusions

The purpose of this paper is to understand the difference in the performance of the prototypes under study. It should be noted that devices with very similar layer compositions, contact types, and thicknesses exhibit significantly different sensitivities when exposed to an X-ray source.

To investigate this, we performed an extensive spectroscopic characterization. The results were compared with electrical measurements and simulations. Simulation results indicate that the presence of the c-Si substrate does not significantly affect the overall device behavior, and its contribution is negligible compared to the signal generated within the a-Si:H layer.

Detectors deposited on Kapton exhibit a more amorphous morphology (Raman), likely due to an increased contribution of void defects within the film. The last affirmation is arguably a qualitative statement that requires feature measurement for quantification. This observation aligns with experimental findings of a broadening of the Si2p components. Additionally, on Kapton, we estimate a lower hydrogen content in the form of monohydride.

The measured mobility gap of the Kapton device is higher (1.9 ± 0.1 eV) compared to the c-Si device (1.7 ± 0.1 eV). According to the literature, this result can be related to a higher degree of disorder and voids in Kapton films. From the mobility gap measurements, we also observe that the two devices show different VM offsets, which could reasonably be attributed to the type of top-layer film.

These findings may explain why devices deposited on Kapton display reduced sensitivity despite having similar noise and leakage current levels. Our hypothesis is that this difference arises from the type of substrate used for deposition, the different morphology, and/or the different alignment of the energy levels. We are planning additional measurements to further support our hypotheses.

## Figures and Tables

**Figure 1 nanomaterials-14-01551-f001:**
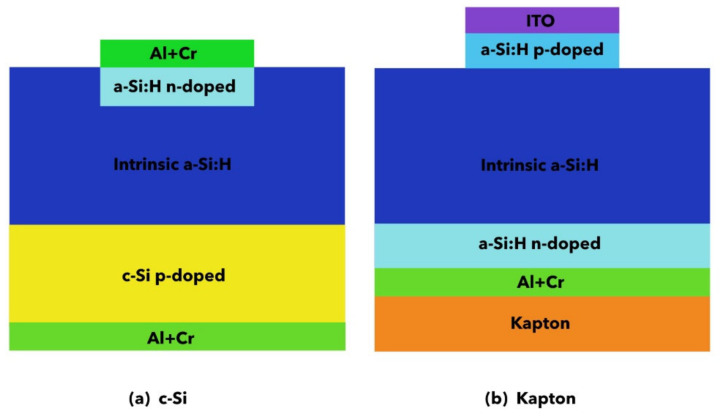
Side view of the p-i-n diodes: (**a**) deposited on crystalline silicon and (**b**) deposited on Kapton. Not to scale.

**Figure 2 nanomaterials-14-01551-f002:**
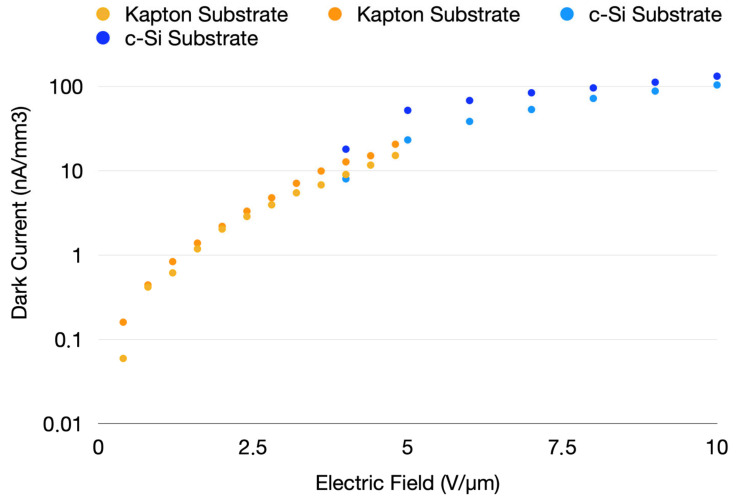
Reverse leakage current comparison for two sensor types: one deposited on c-Si (blue) and the other on Kapton (orange), with leakage normalized to the detector’s sensitive volume.

**Figure 3 nanomaterials-14-01551-f003:**
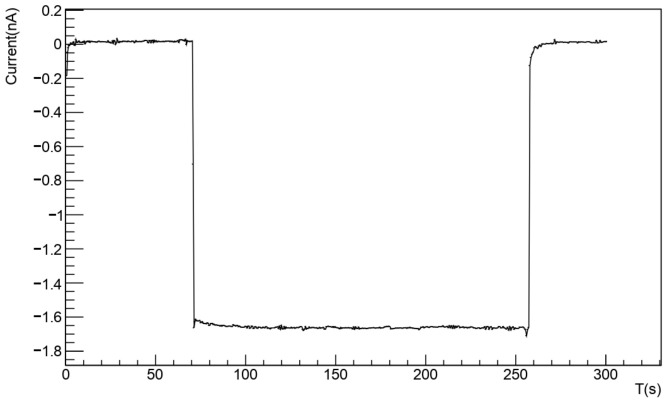
Example of acquired current as a function of time for a known dose rate. The signal is consistently stable over time, with a very low noise level (S/N greater than 10).

**Figure 4 nanomaterials-14-01551-f004:**
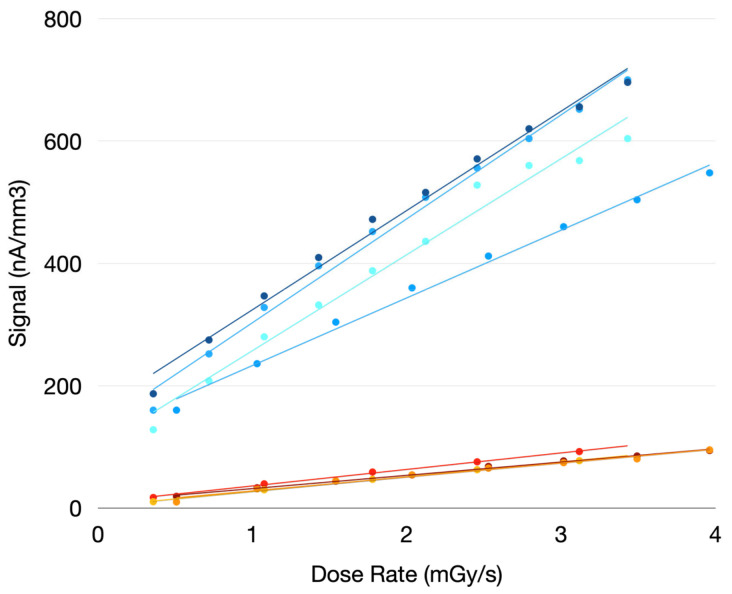
Response curves of the sensor at varying dose rates: blue lines correspond to detectors on crystalline silicon, and orange lines correspond to detectors on Kapton.

**Figure 5 nanomaterials-14-01551-f005:**
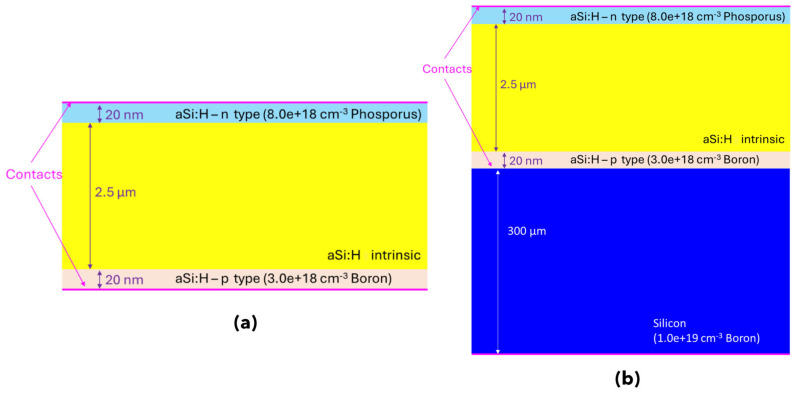
Simulated structure: (**a**) device deposited on Kapton; (**b**) device on p-doped c-Si.

**Figure 6 nanomaterials-14-01551-f006:**
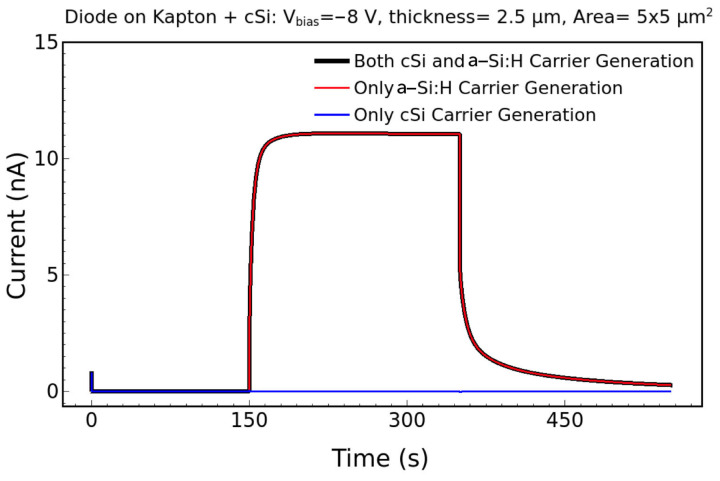
Simulated current–time signal when the carrier is generated in both the a-Si:H and c-Si layers (black), only in a-Si:H (red), and only in c-Si (blue).

**Figure 7 nanomaterials-14-01551-f007:**
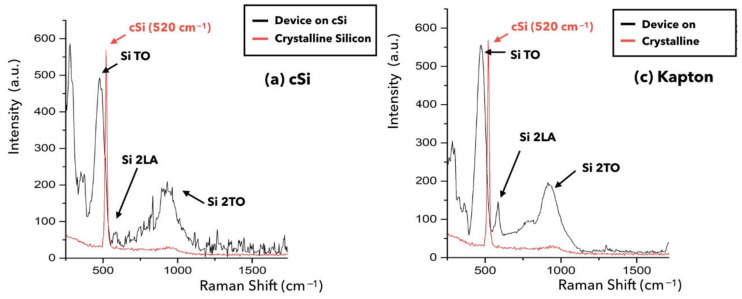
Raman spectra of a-Si:H film on c-Si (**a**) and on Kapton (**b**) substrate. Crystalline silicon spectrum is shown for comparison.

**Figure 8 nanomaterials-14-01551-f008:**
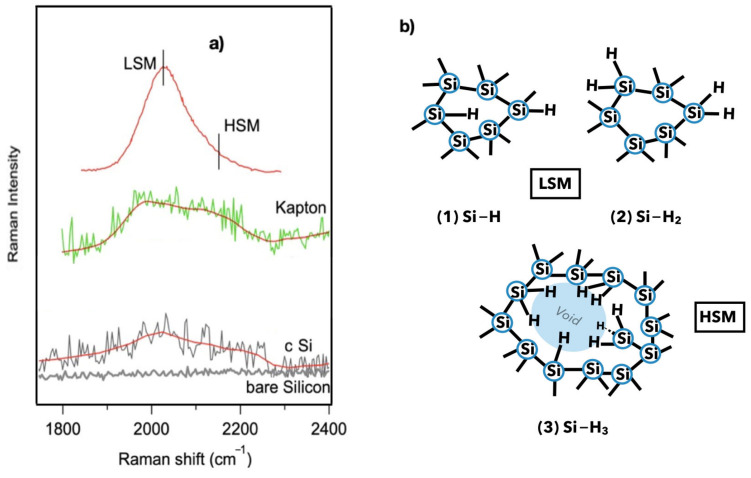
(**a**) Raman spectra in the frequency area of the stretching vibrations of the hydride Si-H and polyhydride bonds of a-Si:H film on crystalline silicon (cSi) and Kapton. The same region in the bare crystalline silicon spectrum is shown for comparison. The smoothed lines are also shown; the top spectrum with the indication of the LSM and HSM is presented for comparison (ref. [22]). (**b**) Schematic representation of (**1**) monohydride, LSM (**2**) dihydride, and (**3**) dangling bonds near the HSM voids.

**Figure 9 nanomaterials-14-01551-f009:**
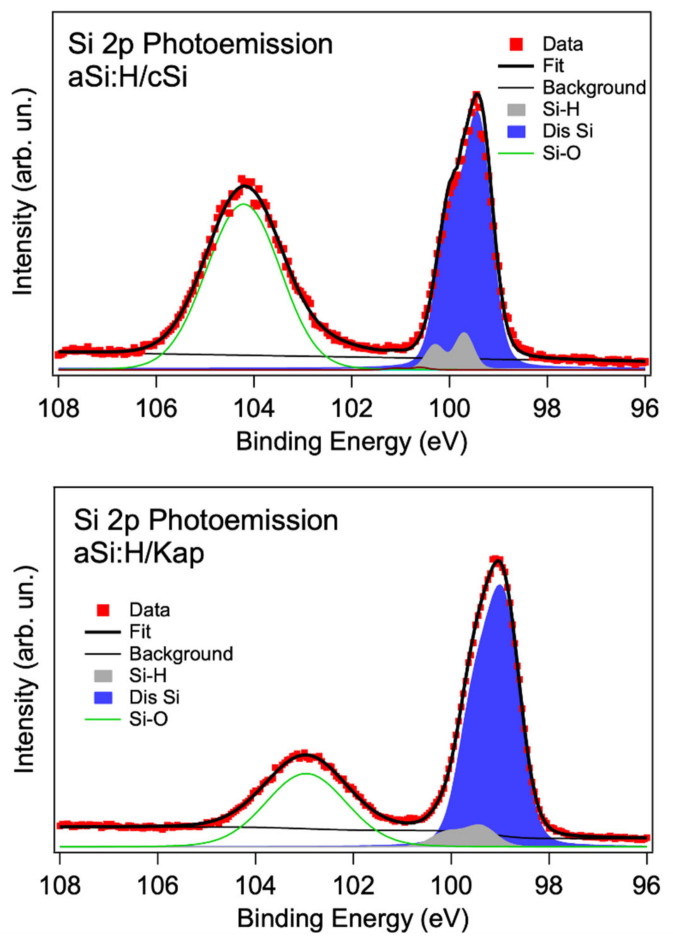
Photoemission Si2p core level deconvolution for a-Si:H device samples: top: a-Si:H/c-Si; bottom: a-Si:H/Kapton.

**Figure 10 nanomaterials-14-01551-f010:**
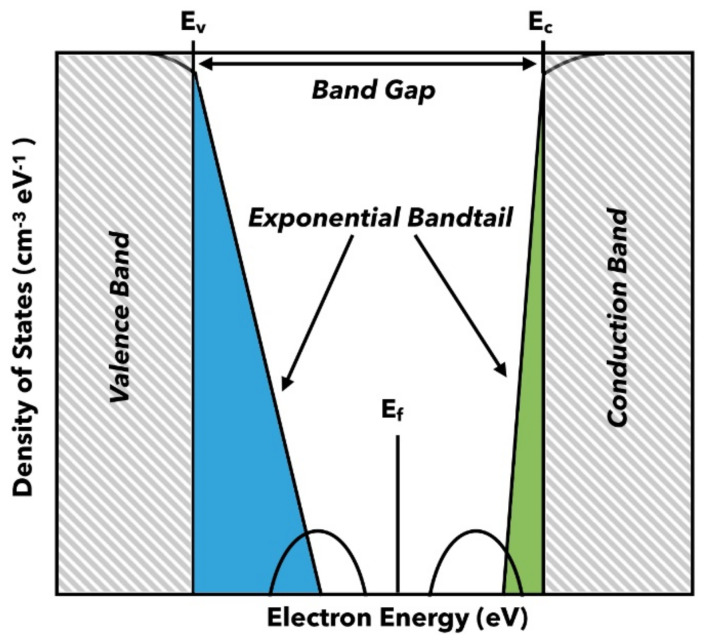
Schematic diagram of distribution of energy states in a-Si:H. Defect states are represented by two equal Gaussian distributions.

**Figure 11 nanomaterials-14-01551-f011:**
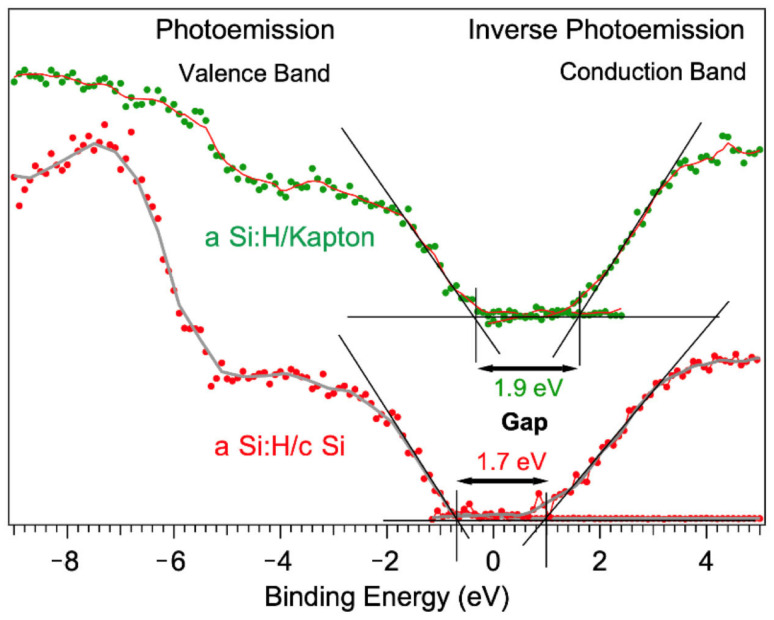
Combined photoemission and inverse photoemission measurements of the Kapton (green) and c-Si samples (red). The lines correspond to the smoothed curves. The value of the transport gap is calculated by the mobility edges of the valence band (photoemission) and conduction band (inverse photoemission) as obtained by the standard graphical method.

**Table 1 nanomaterials-14-01551-t001:** Sensitivity values normalized to volume (and measured with the same electric field of ~2 V/μm) for two types of detector under study. Each value was obtained by averaging the sensitivity of at least 4 sensors with the same characteristics such as substrate and type of contacts.

	n-i-p on c-Si	n-i-p on Kapton
Normalized Sensitivity nC/(cGy mm^3^)	1593.1 ± 60	222.3 ± 6

**Table 2 nanomaterials-14-01551-t002:** Mean values of Gaussian fits obtained by the deconvolution of the acquired Raman spectra, for TO peaks.

Device	TO (cm^−1^)	2LA (cm^−1^)	2TO (cm^−1^)
Kapton	472	570	910
c-Si	481	/	928

## Data Availability

The data presented in this study are available in article.

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
