# Peer review of "Mobility Gaps of Hydrogenated Amorphous Silicon Related to Hydrogen Concentration and Its Influence on Electrical Performance"

_nanomaterials, 2024, doi:10.3390/nano14191551_

Round 1

Reviewer 1 Report

Comments and Suggestions for Authors

In this paper, the characteristics of detectors using amorphous silicon are studied through various analytical methods. The research compares devices with crystalline silicon substrates to those with Kapton substrates. To explain the differences in device characteristics based on the substrate, the study analyzes factors such as amorphous silicon defects, Si-H bonds, and energy bands.

The analysis of device characteristics using methods like TCAD simulation, electrical characterization, Raman spectroscopy, photoemission, and inverse photoemission is comprehensive. However, the figures derived from these analyses do not seem to strongly support the study's conclusions. Specifically, each figure has limitations in directly demonstrating factors such as hydrogen concentration, defect density, and mobility.

  1. It is necessary to explain in more detail how the sensitivity values reported in Table 1 were derived and what these values signify.

  2. Data on the sensor's response to the X-ray dose is needed to explain the sensitivity.

  3. The correlation between the fabricated devices and TCAD simulation results is lacking, particularly in terms of how well the experimental results match the simulations. This needs to be addressed.

  4. The specific modeling parameters and conditions used in the TCAD simulation should be described in more detail. For example, the physical models and their settings used in the simulations should be specified.

  5. Figure 3 shows a V-bias of -8 [V], while the text describes it as 2 [V], which is confusing. Additionally, for precise analysis, data and explanations regarding the input signal over time are needed.

  6. The text mentions that the measurements had a low signal-to-noise ratio, raising concerns about the reliability of the measured data for the device.

Author Response

Thank you very much for all the helpful suggestions and corrections. Below are the responses to your comments.

Comment 1-2:

It is necessary to explain in more detail how the sensitivity values reported in Table 1 were derived and what these values signify.

Data on the sensor's response to the X-ray dose is needed to explain the sensitivity.

Response 1-2: In the revised article, we have provided a more detailed explanation of how sensitivity is calculated and included a reference to our previously published work, which offers an in-depth analysis of signal processing with an X-ray source.

Comment 3: The correlation between the fabricated devices and TCAD simulation results is lacking, particularly in terms of how well the experimental results match the simulations. This needs to be addressed.

Response 3: In this work, we utilized the TCAD suite of tools to investigate only the impact of crystalline silicon (c-Si) on the device's response (layout shown in Fig. 1a) under X-ray exposure. The question we want to answer is:  is it reasonable to expect a contribution due to the presence of crystalline silicon when the device is exposed to X-rays? TCAD is particularly well-suited for simulating c-Si due to its accurate material description by default. And the answer that the simulation software gives us is no. However, the validity of our amorphous silicon (a-Si) modeling approach is confirmed by the good agreement between TCAD simulations and experimental measurements (I-Vs and C-Vs) reported in a recent publication (D. Passeri et al., https://doi.org/10.1016/j.mssp.2023.107870). This allows us to confidently assess the additional contribution of c-Si to the signal. The following sentence has been added in the text to clarify the concept: "One possibility to be understood is if the presence of the c-Si substrate could produce some additional contribution to the signal when X-rays are sent to the device."

Comment 4: The specific modeling parameters and conditions used in the TCAD simulation should be described in more detail. For example, the physical models and their settings used in the simulations should be specified.

Response 4: The specific modeling parameters and activated physics models have been described in the work [D. Passeri et al., https://doi.org/10.1016/j.mssp.2023.107870] therefore the following sentence has been added in the text:"The specific modeling parameters and models accounted for the simulation of a-Si:H devices within the TCAD environment, have been deeply detailed in [15], along with a comprehensive validation against experimental measurements."

Comment 5: Figure 3 shows a V-bias of -8 [V], while the text describes it as 2 [V], which is confusing. Additionally, for precise analysis, data and explanations regarding the input signal over time are needed.

Response 5: We have corrected the error in the text.

Comment 6: The text mentions that the measurements had a low signal-to-noise ratio, raising concerns about the reliability of the measured data for the device.

Response 6: Due to the low signal-to-noise ratio in the Raman spectroscopy measurements, obtaining quantitative results on the hydrogen content within the films is challenging. However, the data is still sufficient to discuss the position of the typical amorphous peak, which is crucial for evaluating the degree of disorder in the film.

Reviewer 2 Report

Comments and Suggestions for Authors

The authors shloud improve this paper.

Comments on the Quality of English Language

My opinion this paper needs a important improvement.

Author Response

Thank you for all the valuable suggestions and corrections. Below, we provide responses to the various comments.

Comment 1: Authors should present figures and tables well. 

Response 1: We have made efforts to improve the presentation of the data and the analysis procedure.

Comment 2: I find the summary and conclusion sections too long. The authors should improve both sections. 

Response 2: We have made an effort to be clearer and more concise.

Comment 3: The abstract section does not explain the results obtained from the article. 

Response 3: We have implemented the modification.

Comment 4: Authors must demonstrate the originality of their work as well as their applications.

Response 4: We have emphasized in the text that the use of this type of detector for measuring particle fluxes other than X-rays has never been successfully carried out. Additionally, the measurement of the band-gap for this type of material in real device has never been performed, nor has any attempt been made to correlate experimental spectroscopic properties with detector performance. 

Comment 5-6-7: 

The question is that the a-Si material is not stable. How can they realize sensitive detectors especially for medical applications? 

They can make detectors based on this kind of material but they will not be reliable and sensitive.

I suggest that authors carefully comment, explain and justify their results. 

Response 5-6-7: This material has been extensively used in flat panel detectors for radiographic measurements since the 1980s and for various other applications. The text now includes a reference to a publication from our collaboration where this type of detector has been successfully employed.

Comment 8: Generally, nanomaterials are used to create ultra-fast, compact and hyper-sensitive detectors. 

Response 8: While generally, nanomaterials are used to create ultra-fast and hyper-sensitive device  in our case, we are not focused on ultra-fast detectors but rather on a detector with high radiation resistance that can be deposited on various types of substrates, including flexible ones, while maintaining uniform performance in very small volumes. In this regard, the chosen material is well-suited to our needs.

Round 2

Reviewer 1 Report

Comments and Suggestions for Authors

No comments